# MACHINE TRANSLATION BASELINES FOR ARABIC- SWAHILI

**Asim Osman , Ahmed Almahdy**
Department of Electrical and Electronics Engineering
University of Khartoum
Khartoum, Sudan
`{asim.awad98,ahmefimadeldin}@gmail.com`

**Muhammed Saeed**
Universität des Saarlandes
`musa00001@uni-saarland.de`

**Hiba Hassan**
University of Khartoum
Khartoum, Sudan
`hiba_im@uofk.edu`

## ABSTRACT

Building neural machine translation (NMT) systems for low-resource languages poses several challenges, mainly due to the lack of parallel data. In this research, we propose a baseline NMT system for translating between Arabic and Swahili. Despite being spoken by nearly 300 million individuals worldwide, the parallel corpus between these two languages is severely underrepresented. To address this, we scraped and processed the largest high-quality parallel corpus of Swahili and Arabic to our knowledge. We then used state-of-the-art NMT models, including Transformers and multilingual variants of Transformers, to build a baseline for bidirectional Arabic-Swahili NMT.

## 1 INTRODUCTION AND RELATED WORK

This paper presents a baseline study for translation between Swahili and Arabic. We utilize the latest state-of-the-art transformer-based models, setting a solid foundation for future research in this domain.

Our experiments show that transfer learning through fine-tuning larger and richer models is effective in improving the translation quality of low-resource language pairs. Specifically, we explore different pre-trained multilingual models such as M2M100 (Fan et al., 2020) and MT5 (Xue et al., 2020) and evaluate their performance for the task of Swahili-Arabic translation. We report our results in terms of BLEU (Papineni et al., 2002) scores. Our best-performing model achieved a BLEU score of 24.47 on the Arabic-Swahili test set, which is a significant improvement over our baseline Transformer model.

Some work has been done on developing neural machine translation baselines for African languages (Ahia & Ogueji, 2020). However, there is no known work for this language pair. And we intend to lay the groundwork for future research.

### 1.1 OUR CONTRIBUTION

1. We release the largest dataset for the Arabic-Swahili language pair collected by processing the Bible and aligning the sentences at the verse level.

2. We validate that using a small dataset to fine-tune a large model improves performance significantly for African languages even for a morphologically rich language like Arabic.

## 2 METHODOLOGY

Three baseline models were trained using the Transformer architecture of Vaswani et al. (2017). we adopted Fairseq (Ott et al., 2019) to train an encoder-decoder transformer model from scratch uti-

lizing sub-word tokenization (BPE) (Sennrich et al., 2015) method to tackle the limited vocabulary issue. the configuration is based on Vaswani et al. (2017) default parameters with a vocabulary size of 12000. and it was trained for 450 epochs.

we conduct fine-tuning experiments with the pre-trained models : mT5 (Xue et al., 2020) with a vocabulary size of 128k and M2M100 (Fan et al., 2020) using Fairseq (Ott et al., 2019) and Huggingface (Wolf et al., 2020) platform.

We trained both models with the same parameters to allow for a fair comparison.

# 3    DATASET

The construction of a parallel corpus[1] is very challenging and requires extensive human expertise. We mapped each verse from the Holy Bible [2] in Arabic to its corresponding verse in Swahili. In doing so, we obtained over 30,000 parallel sentences. We ran a sanity check(Arabic to Swahili), and the Bleu score was 98.4, indicating that the data is aligned. We include the NTREX (Federmann et al., 2022) (News Test Reference for Evaluating MT Models) data set, which is a 2000-sentence parallel corpus.

# 4    RESULTS

All models were trained using BPE (Sennrich et al., 2015). And the fine-tuned models outperform the baseline transformer significantly.

The table below shows our results obtained after conducting several experiments.

Table 1: Test Blue Scores

| Direction | Transformer-base | M2M100 | mt5 |
|---|---|---|---|
| Arabic to Swahili | 10.51 | 24.47 | 15.13 |
| Swahili to Arabic | 11.59 | 10.63 | 7.64 |

# 5    CONCLUSION AND FUTURE WORK

In conclusion, this paper has presented a baseline for low-resource machine translation between Arabic and Swahili. Despite the challenges of working with a low-resource language pair, we have shown that it is possible to achieve reasonable translation quality with the right combination of pre-processing techniques and neural machine translation models. We hope that our results will inspire further research in this area and help to improve access to information and services for Arabic and Swahili speakers, particularly in the context of the African Union's efforts to promote linguistic diversity and inclusivity.

In the future, we plan to collect more parallel data and experiment with different denoising techniques to improve the performance of our models. Also, we want to discover the effect of tweaking the base transformer architecture to better fit the purpose of low-resource translation in addition to get results from back-translation.

URM STATEMENT

We acknowledge that all authors of this work meet the URM criteria of ICLR 2023 Tiny Papers Track.

---

[1]https://github.com/Asimawad/Arabic-to-Swahili-Machine-Translation
[2]http://bible.org

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

## A   APPENDIX

Below are some translation examples produced by our fine-tuned M2M model.

Table 2: Examples

| Description | text |
|---|---|
| Reference | naye bwana akaniambia, nimewaona watu hawa nao ni watu wenye shingo ngumu sana |
| Model Translation | kisha bwana akaniambia, nimewaona watu hawa, nao watu hawa ni watu wenye shingo ngumu |
| Reference | hapo mwanzo alikuwako neno, huyo neno alikuwa pamoja na mungu, naye neno alikuwa mungu. |
| Model Translation | hapo mwanzo neno lilikuwepo, neno lilikuwepo pamoja na mungu na neno lilikuwepo pamoja na mungu. |
| Reference | yesu akajibu, rudini mkamwambie yohana yale mnayosikia na kuyaona |
| Model Translation | yesu akawajibu, nendeni mkamwambie yohana yale mliyoyasikia na kuyaona |
| Reference | hesabu ya waume wote wenye umri wa mwaka mmoja au zaidi walio-hesabiwa walikuwa 6,200. |
| Model Translation | idadi ya wanaume wote wenye umri wa mwezi mmoja au zaidi wa-likuwa 6,200 |

