# OpenReview forum: "MACHINE TRANSLATION BASELINES FOR ARABIC - SWAHILI"
_ICLR.cc/2023/TinyPapers — Submitted to Tiny Papers @ ICLR 2023_

### Official Review · Reviewer_A3ns · 2023-03-27

**Confidence:** 5

**Summary Of Contributions:**

A Baseline NMT system for translating between Arabic and Swahili is presented. A parallel corpus of Swahili and Arabic is scratched and used to build a baseline for bidirectional Arabic-Swahili NMT.

**Rating:**

Great Start (GS): a submission which meets some of the reviewing criteria but has room for improvement

**Strengths And Weaknesses:**


This is a critical topic, and the potential of the open-source parallel corpus is valuable for future research.

The authors should proofread before submitting any manuscript. For example, “full stops”. If you have a full stop, the consecutive sentence should be capitalised.

Table 1 is incomplete. There are no results provided for back-translation, so it is hard to relate to the findings claimed by the authors.

Also, if one of the contributions is “the release of the parallel dataset”, it is vital to mention where and how this dataset can be accessed. It’s not apparent what this dataset will look like or how other researchers can access it.

The basic requirements of the “Tiny” paper are also met. But given the information missing from the paper, the clarity, reproducibility, and correctness are extremely hard to follow (or confirm).

This paper can be published with all the above-mentioned information. But as it is, this is not quite there yet. Authors can get some help to polish their writing and presentation.


**Suggested Changes:**

Suggest reading the review above very carefully. There is a lot of work needed to make this paper into a good high-quality paper. The missing information and results are vital. This paper can be published, provided the authors take the suggestions and resubmit the edited manuscript.

---

### Official Review · Reviewer_fKq9 · 2023-03-28

**Confidence:** 5

**Summary Of Contributions:**

This work creates a bitext for Arabic-Swahili and run a baseline training of a translation model.

**Rating:**

Clear, Correct, and Reproducible (CCR): a submission which meets the reviewing criteria

**Strengths And Weaknesses:**

Creating the parallel corpora for Arabic-Swahili is a valuable resource and it's a nice step into addressing the challenge of translation quality in low-resource languages.

The authors mention that " We ran a sanitycheck, and the blue score was 98.4, indicating that the data is aligned.". What exactly was used for target and reference to compute this bleu score and run the sanity check?

Table 1: Back-Translation column is empty even though in the abstract the autors mentions an improvement when using back translation. I would suggest either remove it or add the numbers if the experiments finish for camera-ready.

Section 4.2 Qualitative: the title is not descriptive enough. The text is very short. While it's good to know the authors looked at the data and deem the quality acceptable, without more analysis it's not helpful for the readers. I would suggest to remove this section and keep the examples in the appendix.

Table 2: the source sentence is missing in the table.

**Suggested Changes:**

- Original backtranslation paper was not cited. I recommend adding this citation since it's one of the main contributions of this work.

- Some information is missing for instance in both tables (mentioned in the previous section).

- Typo: Holy Bible1 -> Holy Bible

- Typo:  blue score -> bleu score

---

### Meta-Review · Area_Chair_96XG · 2023-04-05

**Recommendation:** Invite to archive
**Confidence:** 5

**Metareview:**

**Summary**
* Collected a parallel corpus of Swahili and Arabic texts and built a baseline for bidirectional Arabic-Swahili Neural Machine Translation.

**Strengths**
* Creating a parallel corpus for such low-resourced languages is a very challenging and valuable task for future research.

**Weakness**
* Table 1 is incomplete. The results for back-translation are missing. Hence, it’s challenging to relate to the results claimed by the authors.
* One of the main contributions of this work is the dataset. However, the dataset is not shared. Which hinders the reproducibility of the work.


**Summary:**

Collected a parallel corpus of Swahili and Arabic texts and built a baseline for bidirectional Arabic-Swahili Neural Machine Translation. Strength: Creating a parallel corpus for such low-resourced languages is a very challenging and valuable task for future research.. Weakness: One of the main contributions of this work is the dataset. However, the dataset is not shared.

**Comments And Feedback To The Authors:**

The work is highly relevant. However, the adding the missing details would be super helpful for the research community.

**Reason For Not Giving A Higher Recommendation:**

* The paper has missing results. The authors claimed that the back translation method improved NMT. However, these results are not reported.
* The major contribution of the paper is creating the Arabic-Swahili corpus. However there's no link to that data. This would hinder the reproducibility criteria to fulfil CCR.

Hence the authors need to revise those before the paper to be presented.


**Reason For Not Giving A Lower Recommendation:**

N/A

---

### Decision · Program_Chairs · 2023-04-08

Invite to archive